# In Situ Conservation of Orchidaceae Diversity in the Intercontinental Biosphere Reserve of the Mediterranean (Moroccan Part)

**DOI:** 10.3390/plants14081254

**Published:** 2025-04-20

**Authors:** Yahya El Karmoudi, Nikos Krigas, Brahim Chergui El Hemiani, Abdelmajid Khabbach, Mohamed Libiad

**Affiliations:** 1Ecology, Systematics and Biodiversity Conservation Laboratory, URL-CNRST N° 18, FS, Abdelmalek Essaadi University, M’Hannech II, Tetouan 93002, Morocco; yahyaelkarmoudi@gmail.com (Y.E.K.); b.cherguielhemiani@uae.ac.ma (B.C.E.H.); 2Institute of Plant Breeding and Genetic Resources, Hellenic Agricultural Organisation Demeter (ELGO-DIMITRA), 57001 Thessaloniki, Greece; 3Department of Viticulture, Floriculture & Plant Protection, Institute of Olive Tree, Subtropical Crops and Viticulture, Hellenic Agricultural Organization Demeter (ELGO-DIMITRA), 71307 Heraklion, Greece; 4Biotechnology, Environment, Agri-Food and Health Laboratory, Faculty of Sciences Dhar El Mahraz, Sidi Mohamed Ben Abdellah University, Fès 30003, Morocco; khamajid@hotmail.com

**Keywords:** *Cannabis* farming, forest fires, human activities, orchids, parks, threatened species, water drainage

## Abstract

The focus of this study was the Intercontinental Biosphere Reserve of the Mediterranean (IBRM, part of the biodiversity hotspot of the Mediterranean Basin) and the Orchidaceae family, which is under-studied in the Moroccan part of the IBRM. For this reason, an inventory of Orchidaceae diversity and factors that could influence their in situ conservation was undertaken, employing a series of field surveys conducted in the Northern Moroccan IBRM ecosystems. In total, 42 sites were surveyed in four protected areas of the Moroccan part of the IBRM. In total, 21 Orchidaceae species and subspecies (taxa) belonging to seven genera were identified, including *Orchis spitzelii* subsp. *cazorlensis,* as newly recorded in Morocco, as well as several new reports for different sites and/or areas surveyed, thus updating the previous knowledge of Moroccan Orchidaceae. Most of the Orchidaceae taxa were found in limited numbers of individuals (<30) and were restricted in a few sites (1–3) or a single area; thus, they were assessed as poorly conserved due to the scarcity of rainfall coupled with human pressures, such as the abstraction of surface water, forest fires, and the conversion of protected forests to *Cannabis* farms. The enforcement of existing laws, the adoption of strategies to combat desertification and forest fires, the prohibition of *Cannabis* farming, and raising awareness among the local population could reduce the pressures on the protected Orchidaceae members and their habitats, thereby contributing to their conservation.

## 1. Introduction

Orchidaceae is one of the largest and most diverse botanical families of vascular flowering plants, with over 29,524 accepted species in 736 genera [1,2,3]. Wild orchids can grow almost worldwide, from tropical rainforests to arctic tundra, though most species are found in tropical regions colonizing diverse habitats (terrestrial, epiphytic, and/or lithophytic). In North Africa and Europe, orchids are exclusively terrestrial, inhabiting different types of ecosystems such as forests, scrublands, meadows, moors, peat bogs, marshes, etc. [4,5,6,7].

Several studies have reported that species richness, distribution, abundance, and growth and the reproduction of terrestrial orchids are dependent on several biotic and abiotic factors, such as pollinator insects, vegetation types, geological substrates, soil proprieties, climate, latitude, longitude, elevation, exposure, and temperature [6,7,8,9,10]. Previous studies from North Africa have reported on the richness and diversity of Orchidaceae members, their distribution, and their in situ conservation [11,12,13]. These works have highlighted the large diversity of Orchidaceae in North Africa and emphasized the need for additional prospection and protection regarding certain taxa and areas [11,12,13]. Despite their ecological importance, Orchidaceae members of the Mediterranean basin and their habitats are threatened by several human activities, namely annual crops, fire, grazing, over-collection, recreational activities, and urbanization [6,14,15,16,17]. Moreover, previous studies ring the bell that the presence of more than 135 Orchidaceae taxa (species and subspecies) is threatened by legal and/or illegal trade [17,18,19,20,21,22].

Globally, the family Orchidaceae has been protected since 1975 by the Convention on International Trade in Endangered Species of Wild Fauna and Flora (CITES) Appendices I and II [23]. However, according to the International Union for Conservation of Nature (IUCN) Red List [22], only 271, 479, 245, and 108 taxa of Orchidaceae are assessed as Critically Endangered (CR), Endangered (EN), Vulnerable (VU) and Near Threatened (NT), respectively, while six taxa are considered Extinct (Ex). Regionally, 20 orchids of Spain are included in the IUCN Red List, including one threatened (EN) and three Near Threatened (NT) taxa; in contrast, however, only 13 taxa are included in the IUCN Red List, and seven taxa are assessed as threatened in North Africa. In Morocco, only three taxa, namely *Dactylorhiza maurusia* (Emb. & Maire) Holub (EN), *Dactylorhiza elata* (Poir.) Soó (NT), and *Platanthera algeriensis* Batt. & Trab. (NT) are assessed as threatened [22,23]. The latter two orchid species share the same threat categories in Morocco and Spain. Following global trends, the Orchidaceae members in Morocco have been recently designated with a special protection status governed by the National Law No. 29-05 of 2011 on the protection of species of wild fauna and flora and their trade [24]. Given that the legally protected areas, such as nature reserves and parks, could constitute an effective tool for the in situ conservation of globally threatened biological resources, Morocco has designated four biosphere reserves, ten national parks, and one natural park to protect its biodiversity in situ [25].

Globally, there are more than 331 studies focusing on orchids in several protected areas of the world [26], while there are only a few studies concerning the Orchidaceae members in various parts of Morocco [14], the Bouhachem Natural Park [27], or the Talassemtane National Park [28,29]. All previous studies and taxonomic revisions taken together report the presence of only 54 taxa of Orchidaceae in Morocco [27,28,29,30,31]. Despite the general legal protection measures taken by biodiversity managers in Morocco, the current threats to Orchidaceae members and their habitats persist. Nonetheless, severe knowledge gaps hinder the protection of Orchidaceae members in Morocco as compared to neighboring countries that have conducted large inventories of Orchidaceae members, e.g., in the Iberian Peninsula and the Balearic Islands with more than 130 taxa [32,33], in Spain with 96 taxa [15], in Northeast Algeria with 64 taxa [12], or in Tunisia with 50 orchid taxa [11]. Moreover, Spanish managers have gone further by including 76% of the domestic native orchids in some catalogue of legal protection, such as the IUCN Red List, the catalogue of regional protection, or the regional Spanish red list [15]. In Andalusia (Southern Spain), including the northern part of the IBRM, four Orchidaceae taxa are included in the regional catalogue (RC) of Andalusia (DEC 23/2012) [15]. These conservation efforts have led to a reduction in threats to Spain’s orchids. This situation has facilitated the use of orchids as bioindicators in habitat management and species conservation [15,34]. The situation in Algeria is no better than in Morocco, with only seven taxa protected under the Algerian law for the conservation of wild plant species of 2012 [35]. Due to the gaps in knowledge regarding the diversity and threats of Orchidaceae compared to neighboring countries, Morocco and other North African countries must accelerate the inventory speed regarding the conservation of their Orchidaceae members, especially in the IBRM region, to prevent their possible disappearance in the face of the ongoing biodiversity loss crisis. To this end and due to conservation concerns, the present investigation firstly aimed to provide an inventory of the Orchidaceae members in the protected areas at the southern range of the Intercontinental Biosphere Reserve of the Mediterranean (IBRM) extending to Northern Morocco, and secondly, to identify the conservation challenges, including man-made threats, faced by them within the study area. The following questions have driven our survey: What is the diversity of orchids naturally found in the protected areas of the IBRM? In what habitats do these orchids grow naturally in the IBRM? How big is the current population per recorded orchid taxon in the IBRM protected areas, and what man-made pressures/threats are faced locally? What distribution patterns can be detected among the different orchid taxa recorded? The outcome of this inventory is aimed to serve as a baseline for future monitoring programs implementing conservation strategies.

## 2. Materials and Methods

### 2.1. Study Area

The Intercontinental Biosphere Reserve of the Mediterranean (IBRM) is situated in Northern Morocco between latitudes 34.793° N and 35.927° N and longitudes −5.842° W and −5.036° W (Figure 1), extending to Andalusia in Spain. The IBRM was established in 2006 under the UNESCO Man and Biosphere Reserve Program as an intercontinental cross-border region, extending over an area of almost one million hectares in total and shared almost equally between the Moroccan and Spanish territories [25,36]. Both IBRM parts, namely Northern Morocco (Rif region) and Southern Spain (Andalusia), are ecosystem-rich and biodiversity-rich, including Mediterranean forests, coastal ecosystems, high-altitude mountain areas, and riparian zones, thus supporting diversified flora and fauna with a high rate of endemism. Only 30% of the territory is protected in the Moroccan part, compared to 70% in the Spanish part [36]. The Moroccan part of IBRM includes the mountainous region of the Western Rif. It covers a large part of the Province of Chefchaouen and various areas of the Wilaya of Tetouan and the Provinces of Fnidek, Fahs-Anjra, and Larache. This part of the IBRM contains many areas of biological and ecological interest, e.g., the Talassemtane National Park, the Bouhachem Natural Park, the Jbel Lahbib Reserve, and the Dardara area [25,36], in which we recorded members of Orchidaceae. Other surveyed protected areas of IBRM were not included in this study due to the absence of detected Orchidaceae members, namely Ben Karich Forest, Smir Lagoon, and Jbel Kelti.

The climate of the study area is generally temperate, with dry and warm summers in the mountains of Talassemtane National Park and Bouhachem Natural Park, and temperate, dry, and hot summers in the remaining territory [37]. The annual precipitation may vary from 545 mm to 1209 mm (mean = 892 mm). However, the annual mean temperature may range from 8.85 °C to 18.79 °C (mean = 16.46 °C) (data extracted from https://www.worldclim.org/ (accessed on 28 August 2023) [38]).

From a geological viewpoint, the study area is divided into two parts. The first part is composed of the Talassemtane National Park, the Dardara area, and the Jbel Lahbib area with a limestone-dolomite bedrock. The second part is composed of the Bouhachem Natural Park, which is essentially made up of acidic layers of flysch and sandstone. The most important mountains in the study area are Jbel Kelti (1821 m a.s.l.), Jbel Lakraa (2151 m a.s.l.), Jbel Tissouka (2122 m a.s.l.), and Jbel Bouhachem (1658 m a.s.l.) [29,39,40].

### 2.2. Field Surveys

To study the diversity, distribution, and conservation status of Orchidaceae of the protected areas of the IBRM of Northern Morocco, we followed a previously published methodology for comparison reasons [14]. In brief, different types of natural and man-impacted ecosystems were visited repeatedly (1 May 2019 to 18 May 2024), surveying 42 sites at different altitudes, ranging from 18 m to 2147 m above sea level (a.s.l.), and covering different ecosystems (forest, scrubland, lawn, wetlands, etc.) [27,28,29] on several occasions. The surveys were carried out randomly by walking through the study area. Once we encountered an Orchidaceae member on a site, we proceeded to study the diversity of extant orchids and the ecological factors that could influence their diversity and conservation, as previously described in similar studies of nearby regions of the IBRM [14].

To document the occurrence and conservation challenges related to local Orchidaceae members, based on previous studies [8,10,41,42,43], we selected several parameters to be recorded during the field surveys, such as in situ species richness, relative abundance, plant communities, coverage rate of vascular vegetation, altitude, GPS coordinates (latitude and longitude), exposure, slope, substrate, moisture, and perceived human impacts (threat/pressure types), as previously described in similar studies of nearby regions of the IBRM [14].

During the field study, orchid richness was determined in situ by counting the number of detected orchid species and subspecies, as well as the number of co-occurring vascular plant species of other families. The abundance of orchids (number of orchid individuals per site) was determined by counting the total number of orchid specimens in each site [14]. The coverage rate of vascular plants was estimated by assigning standard coverage percentages, namely 5% (rare occurrence), 5–25% (very sparse), 25–50% (sparse), 50–75% (dense), and 75–100% (very dense occurrence) [41]. Plant communities associated with orchids were assessed based on their physiognomy and were grouped into 14 habitat types based on field notes (Table 1). Soil moisture was assessed in the field by roughly estimating the amount of water in the soil, and three basic moisture classes were designated, i.e., dry, moderate, and moist habitats [41]. The geological bedrock types were determined based on the geological map of the Rif (Northern Morocco) [39].

Altitude was measured using a hand-held smartphone with a global positioning system (GPS). Slope was estimated visually as the percentage of the Earth’s tilt relative to the vertical axis. The global positioning system coordinates (longitude, latitude) were recorded for each site studied. Later, the GPS coordinates were used in ArcGIS (10.8) software to prepare a distribution map of the surveyed sites, illustrating the matrix for the spatial distribution patterns of the recorded orchid taxa. Exposure was additionally determined and was used to define the orchids’ growth.

To determine the level of in situ conservation of Orchidaceae members in the Moroccan part of the IBRM, we noted different extant threats/pressures according to the IUCN human activity categories [44] in the study area that were likely to threaten the orchid populations.

In each site studied, the living (fresh) specimens of Orchidaceae species were identified in situ and photographed without harming the wild-growing individuals and populations. Flowers of each taxon were collected for taxonomic and nomenclature validation based on the available floras [30,31] and the specific work of Delforge [6], whereas the nomenclature followed the Plants of the World online database [45]. The specimens of orchids were deposited in the herbarium of the Laboratory of Ecology, Systematics and Biodiversity Conservation of the Faculty of Sciences Tetouan, Abdelmalek Essaadi University, Morocco.

## 3. Results

### 3.1. Orchid Richness in Sites

In total, 42 sites were studied in the Moroccan part of the Intercontinental Biosphere Reserve of the Mediterranean (Figure 1; Table 2), spanning from lowlands (77–104 m a.s.l. in the Jbel Lahbib Reserve) to mountainous areas (330–1293 m a.s.l. in the Bouhachem Natural Park or 415–1969 m a.s.l. in the Talassemtane National Park). The richest sites of orchid taxa were 12 and 13 of the Talassemtane National Park and 41 of the Jbel Lahbib Reserve, with six, five, and five taxa, respectively (Table 2). These sites provided habitats of matorral with *Quercus ilex* L. (site 13), rainforest with *Abies pinsapo* subsp. *marocana* (Trab.) Emb. & Maire (site 12) and matorral with *Pistacia lentiscus* L. and *Myrtus communis* L. (site 41). Overall, a total of 1138 orchid individuals of all taxa were recorded in the studied Moroccan part of the IBRM, and the richest sites for numbers of orchid individuals were 34 (Dardara) and 7 and 12 (Talassemtane National Park), which provided habitats for 120, 80, and 73 orchid individuals, respectively (Table 2).

From a geographical point of view, the Talassemtane National Park was the richest, with 17 Orchidaceae taxa, compared to Bouhachem Natural Park with only 7 taxa, or the Dardara area and the Jbel Lahbib Reserve with only 6 taxa.

### 3.2. Species Richness, Distribution, and Abundance

This floristic inventory reports the taxonomic identification of 21 Orchidaceae taxa (species and subspecies) or 26 taxa according to the African Plant Database (APD) [46] in the studied area (see Appendix A), which belong to seven genera (*Cephalanthera, Epipactis, Himantoglossum, Limodorum, Ophrys, Orchis,* and *Serapias*). In total, nine orchid taxa were recorded in only one of the four areas studied (Figure 2), ten taxa were found in two of the studied protected areas (Figure 3), and only *Serapias lingua* L. subsp. *lingua* and *S. parviflora* Parl. were found in three or all the studied areas (Figure 4), respectively.

The genus *Ophrys* was the most abundant in the study area, with seven taxa (Table 3). The discrepancy on the exact number of recorded *Ophrys* taxa (cases marked with asterisks in the floristic catalogue of Appendix A) concerns the following complex and unresolved taxonomic-nomenclatural issues: (i) *Ophrys* x *numida* Devillers-Tersch. & Devillers and *Ophrys* × *battandieri* E.G.Camus—although distinct species of hybrid origin according to APD [46], they are synonymized under the latter according to POWO [45]; (ii) *Ophrys sicula* Tineo (present in Tunisia and Libya, not Morocco) according to APD [46] and *Ophrys lutea* Cav. subsp. *galilaea* (H.Fleischm. & Bornm.) Soó—although synonymous taxa according to POWO [45], they are considered distinct taxa according to APD [46]; (iii) *Ophrys flammeola* P.Delforge and *Ophrys fusca* Link subsp. *fusca*—although distinct taxa according to APD [46], they are synonymized under the latter according to POWO [45], and (iv) no subspecies is undoubtfully distinguished in *Ophrys tenthredinifera* Willd. according to POWO [45]; however, the APD [46] recognizes three geographical subspecies as morphologically distinct, namely subsp. *tenthredinifera,* subsp. *ficalhoana* (J.A.Guim.) M.R.Lowe & D.Tyteca, and subsp. *grandiflora* (Ten.) Kreutz (Figure 5). See also Appendix A for the local occurrences in the study area of the three distinct subspecies according to APD [46].

*Orchis mascula* (L.) L. subsp. *laxifloriformis* Rivas Goday & B.Rodr. and *Ophrys tenthredinifera* were the most frequently recorded taxa with 156 and 151 counted individuals, respectively, occurring in nine sites; these were followed by *Ophrys* x *battandieri, Neotinea maculata* (Desf.) Stearn, and *Cephalanthera longifolia* (L.) Fritsch, which were growing in six sites with 69, 63, and 54 individuals, respectively (Table 3). Most of the recorded Orchidaceae (10 out of 21 taxa) were restricted to two of the studied areas (Figure 3); 9 out of 21 taxa were found in one of the studied areas (Figure 1), and only *Serapias lingua* subsp. *lingua* and *Serapias parviflora* were found in three or four studied areas (Table 3, Figure 4). For example, *Orchis spitzelii* Saut. ex W.D.J.Koch subsp. *cazorlensis* (Lacaita) D.Rivera & Lopez Velez and *Serapias vomeracea* (Burm.f.) Briq. were found only in a specific area or site, an occurrence with limited numbers of individuals (<15), while another six taxa (*Himantoglossum hircinum* (L.) Spreng., *Ophrys fusca* subsp. *fusca*, *Ophrys lutea* subsp. *galilaea*, *Ophrys scolopax* Cav. subsp. *apiformis* (Desf.) Maire & Weiller, *Ophrys speculum* Link, *Serapias strictiflora* Welw. ex Veiga) were only found in two or three sites located in one or two of the studied areas, with a limited number of individuals (32 or fewer) (Table 3). Overall, 11 out of 21 Orchidaceae taxa recorded in the Moroccan IBRM were found with 35 or more individuals (Table 3).

In the study area, seven taxa could be considered as prioritized in conservation terms due to narrow distribution ranges combined with a small number of individuals (*Epipactis tremolsii* Pau, *Himantoglossum hircinum*, *Limodorum trabutianum* Batt., *Ophrys fusca* subsp. *fusca, Orchis mascula* (L.) L., *Orchis spitzelii* subsp. *cazorlensis*, and *Serapias vomeracea*), thus requiring in situ conservation monitoring (Table 3). Novelty-wise, this inventory identified one new Orchidaceae taxon for the first time in Morocco, namely *Orchis spitzelii* subsp. *cazorlensis* (Figure 6).

The study area was also found to be rich in non-Orchidaceae vascular plants, with 55 taxa belonging to 44 genera and 30 families of vascular plants (see Appendix A). This part of plant diversity is an important factor affecting the distribution and the abundance of Orchidaceae species in the study area. In total, 14 types of plant communities were defined in this study (Table 1). The Orchidaceae species of the study area were primarily found in rainforests (26 sites), with *Quercus* spp. and/or *Abies pinsapo* subsp. *marocana* at high elevation and rainforest with *Quercus suber* L. at low elevation. However, matorrals with *Quercus ilex*, *Pistacia lentiscus,* and/or *Pistacia atlantica* Desf. (13 sites) also provided a suitable habitat at low elevation. The rainforest habitat with *Quercus ilex* and *Cistus* spp. (habitat type 8) and the matorral with *Quercus ilex* (habitat type 6) were the most suitable habitats for the growing of wild orchids, each distributed in seven sites with 20 and 12 non-Orchidaceae vascular plants, respectively (Figure 7).

### 3.3. Threats/Pressures and In Situ Conservation

In this study, eight types of human threat/pressures were recorded in situ during the Orchidaceae inventory, such as small-holder grazing, ranching or farming, roads, tourism and recreation areas, solid waste, annual and perennial non-timber crops (such as *Cannabis* farming), increase in fire frequency/intensity, gathering terrestrial plants, and abstraction of surface water (agricultural use). No apparent threat type/pressure was recorded in ten sites of the Talassemtane National Park (Table 2). While at least one of these human activities was detected in most of the surveyed sites (*n* = 32), small-holder grazing constituted the most abundant type of threat, followed by roads and tourism and recreation areas, which were present in 20, 18, and 10 sites, respectively (Table 2). In sites 36 and 37 of the Dardara area, we recorded a simultaneous presence of four types of human threats/pressures that could deteriorate the natural habitats of Orchidaceae (small-holder grazing, roads, tourism–recreation areas, and solid waste) (Table 2). In the Talassemtane National Park, a protected area of the IBRM, fire and *Cannabis* farming destroyed a large area of the holm oak and fir forest. In this study, the repeated surveys of 2021 and 2022 evidenced the disappearance from specific sites in the Talassemtane National Park of 12 and 2 extant individuals of *Orchis mascula* and *Himantoglossum robertianum* (Loisel.) P.Delforge, respectively. Before the wildfires, *Cannabis* farming was practiced on a small scale, utilizing local varieties that were less water demanding and locally adapted to the environmental and cultural contexts. During the inventory, we found out that this trend was reversed due to the significant expansion of *Cannabis* farms utilizing new varieties imported from abroad (called locally “Khardoula”) that were more water demanding. Currently, *Cannabis* farming has gradually colonized parts of the lands burned by the devastating intentional fires in recent years.

Another factor threatening biodiversity conservation in the IBRM is the abstraction of surface water coupled with the scarcity of precipitation. A shocking case of the effects of the abstraction of surface water was encountered in a wetland rich in *Serapias* spp., namely site 7 of the Talassemtane National Park. In this site, the surveys conducted in April 2021 and May 2022 revealed a high richness and abundance of *Serapias* spp. However, during the next survey of April 2023, we observed a drying out of the wetland and a very marked reduction in the abundance of *Serapias* spp.

The aforementioned human threats could facilitate biological invasions and/or ruderalization in the study area. For example, we have recorded the presence of the expansive native species *Rubus ulmifolius* Schott in sites 9, 22, and 30, and a species associated with disturbed habitats, namely *Dittrichia viscosa* (L.) Greuter, in site 16. Such entities could pose a threat to orchids and their companion vegetation in Talassemtane National Park and Bouhachem Natural Park.

In addition to human factors, natural processes could also threaten IBRM’s Orchidaceae members, such as widespread rugged relief and consequent erosion. Thus, due to the mountainous nature of the study area, the slope of the terrain where habitats of Orchidaceae occur ranged from 20% to 60%, with 66.9% of the surveyed sites having >40% inclination.

Some of the recorded orchids occurred only in high-altitude wetlands, representing sensitive and important habitats of terrestrial orchids. For example, in the Talassemtane National Park mountains (site 7) and the Bouhachem Natural Park (sites 25, 29, and 30), *Serapias* spp. were found only in high-altitude wetlands. In these wetlands, there was a high abundance of *Serapias* spp. with numerous visiting and/or nesting insects profiting from the flower nectar of *Serapias* spp. in return for their pollination. The latter outlined a specialization of some *Serapias* spp. for occurrence in high-altitude Moroccan wetlands whose disturbance and/or drying out could irreversibly affect these Orchidaceae members and their pollinating insects.

Except for *Serapias vomeracea*, all remaining orchids of the study area (*n =* 20) could grow on moderately humid substrate, and seven taxa could grow on rather humid substrate, namely *Orchis mascula* subsp. *laxifloriformis*, *Orchis mascula*, *Himantoglossum robertianum*, *Serapias lingua* subsp. *lingua*, *Serapias parviflora*, *Serapias strictiflora*, and *Serapias vomeracea*. Only *Ophrys* x *battandieri* could grow on dry substrate.

The Orchidaceae taxa of the study area were found to occupy habitats oriented towards less sunny exposures to avoid soil dryness and evapotranspiration caused by solar radiation. For example, 52.3% of the surveyed sites with Orchidaceae members were exposed to the North (15 sites) and East (9 sites), while 28.6% were exposed to the Northwest (8 sites) and Southeast (4 sites). However, despite the coverage of vascular plant vegetation exceeding 50% in 85.7% of the surveyed sites in the study area (Table 1), the recorded orchids tended to occupy the open spaces and/or the edges of roads and trails to escape from interspecific competition and/or shading by tall vegetation.

## 4. Discussion

The inventory presented herein revealed the occurrence of 21 Orchidaceae taxa (or 26 according to APD [46]) in the Northern Moroccan part of the Intercontinental Biosphere Reserve of the Mediterranean. This richness is less than that of the West Rif area of Morocco [27] or almost equal to the West Bank area of Palestine [16]. The 7–12 *Ophrys* taxa (unresolved taxonomic entities) found in the study area represent about 6% of all the taxa of the genus *Ophrys* in Europe, North Africa, and the Middle East [6]. This genus is also the most diverse in the Mediterranean basin, being endemic to the West Palearctic region [6,47].

Recently, previous studies [48,49] highlighted the heterogeneity of orchid groups and the significant morphological and phenological variation within the *Ophrys tenthredinifera* group. The analysis of the *O. tenthredinifera* group in the study area led to the identification of three subspecies (Figure 1). Although these subspecies are not recognized by standard online databases [45], they are reported as distinct in other online reference databases [46]. The distinction of different subspecies in the *Ophrys tenthredinifera* group for the case herein studied could be of use in wider future studies examining specimens across the geographical range of these subspecies.

Compared to previous studies [14,27,28,29,30], the current inventory in the IBRM has increased the diversity of known Orchidaceae in Morocco by reporting *Orchis spitzelii* subsp. *cazorlensis* as a new floristic record. Currently, this family includes 51 species and subspecies in the Rif region, Morocco. Moreover, our study reported the presence of four additional taxa of Orchidaceae in the Talassemtane National Park compared to previous studies [28,29], namely *Himantoglossum robertianum*, *Orchis mascula*, *Orchis spitzelii* subsp. *cazorlensis,* and *Serapias lingua* subsp. *lingua*. In the case of the Bouhachem Natural Park, this study reported the presence of two new taxa of orchids compared to earlier studies [27], namely *Orchis mascula* and *Serapias vomeracea*. It should also be noted that some orchid species have never been recorded before our study in the Jbel Lahbib or the Dardara areas, thus representing local first-time reports. The Venn diagram across related studies (Figure 8) shows that *Orchis spitzelii* subsp. *cazorlensis* has not been recorded before (new report), and 11 taxa recorded by the floristic catalogue of Valdés et al. [30] were not reported by previous studies in the Rif region [14,27,28,29] or the current study. These taxa are *Anacamptis collina* (Banks & Sol. ex Russell) R.M.Bateman, Pridgeon & M.W.Chase, *Anacamptis coriophora* (L.) R.M.Bateman, Pridgeon & M.W.Chase, *Anacamptis papilionacea* (L.) R.M.Bateman, Pridgeon & M.W.Chase, *Androrchis spitzelii* (Saut. ex W.D.J.Koch) D.Tyteca & E.Klein, *Dactylorhiza romana* (Sebast.) Soó subsp. *romana*, *Limodorum abortivum* (L.) Sw., *Ophrys atlantica* Munby, *Ophrys fusca* subsp. *dyris* (Maire) Soó, *Platanthera algeriensis*, *Serapias cordigera* L., and *Spiranthes spiralis* (L.) Chevall. The situation for *Serapias cordigera* is almost similar to that of Algeria and Tunisia, where this species has not been recorded for a long time [11,12]. If not an artifact of diversified sampling efforts in different studies, this finding could imply the progressive regression of orchid populations in the Moroccan Rif. Undoubtedly, additional studies are needed in the Moroccan Rif, with a particular focus on regions not covered by previous research. Another seven orchid taxa were commonly found in the four previous works [14,27,28,30] and the current study, namely *Cephalanthera longifolia*, *Epipactis tremolsii*, *Ophrys apifera* Huds., *Ophrys lutea* subsp. *galilaea*, *Orchis anthropophora* (L.) All., *Neotinea maculata*, and *Serapias parviflora.*

Despite the fact that *Orchis spitzelii* subsp. *cazorlensis* [=*Orchis cazorlensis*] has been reported as an endemic taxon of the Iberian Peninsula [50] and has been assessed as having a doubtful presence in Morocco according to [31], the study herein confirmed the presence of this species in the Talassemtane National Park of Morocco, albeit with only a few individuals (*n* = 13), thus delivering a first documented report substantiated with herbarium material and ecological observations. This species is characterized by a large, short, and broadly open spur, olive-green to reddish-brown sepals outside, a lip that is not very convex and rather spreading, and a spur ca. half the length of the lip [6].

In the age of climate change and biodiversity crises, habitat loss could generate a decline in the richness and abundance of terrestrial orchids [51]. In our study area, eight types of human threats/pressures were recorded as deteriorating factors for the recorded Orchidaceae populations and their wild habitats. Given that an orchid species is threatened if, among other things, it has a comparatively higher specificity of habitat preferences and a smaller geographic distribution and population size [51,52], our study showed that many of the recorded orchid species occurred in only a low number of sites (only one or two sites) and/or with a limited number of individuals (<10 individuals), such as *Limodorum trabutianum*, *Ophrys fusca* subsp. *fusca*, *Orchis spitzelii* subsp. *cazorlensis,* and *Serapias vomeracea*. Some other recorded Orchidaceae members were threatened by multiple human activities, such as *Orchis mascula* and *Himantoglossum robertianum.* Surprisingly but not unexpectedly, during the surveys of 2021 and 2022, we observed the disappearance of the latter two species from their natural habitats (sites 4 and 3, respectively). Despite the presence of several human activities that could threaten orchids in the study area, only *Limodorum trabutianum* is currently assessed as Near Threatened globally by the IUCN [22]. Regionally, *Limodorum trabutianum* and *Serapias vomeracea* are assessed as Near Threatened and Endangered, respectively [53]; this implies that wider research and assessments on the conservation status of Orchidaceae in Morocco are required to inform the global IUCN checklist of Orchidaceae.

Although most of the recorded Orchidaceae in the studied part of the IBRM were assessed herein as threatened by human activities, special attention and priority should be given to those of calcareous substrates occurring from lowlands to intermediate altitudes due to their proximity to human agglomerations and expansive urbanization. Orchids of such habitats may have a high rate of rapid decline, as reported by previous studies [51] for similar cases in Estonia and the UK or by others [41] in Serbia. However, our study showed that the limestone fir and holm oak forests in intermediate altitudes of the Talassemtane National Park were also highly influenced by human activities, namely the abstraction of surface water, forest fires, and *Cannabis* farming. These areas represent the only occurrences of *Orchis spitzelii* subsp. *cazorlensis* reported herein for the first time in Morocco. Therefore, orchid populations of calcareous habitats in the study area could diminish gradually in the coming years unless in situ conservation measures are quickly implemented.

In the Talassemtane National Park, several of the recorded orchids were found primarily in natural coniferous forests of *Abies pinsapo* subsp. *marocana* and *Pinus nigra* J.F.Arnold subsp. *mauretanica* (Maire & Peyerimh.) Heywood (according to [46] or subsp. *salzmannii* (Dunal) Franco according to [45]), which are local endemics and protected by Law No. 29-05 of 2011 on the protection of species of wild fauna and flora and their trade [24]. Despite the protection status designated for the local endemic conifers, the burning of these forests and their uncontrolled conversion into *Cannabis* farms give rise to major concerns about the conservation of these orchid species and their habitats, as previously argued in other studies [54].

Water drainage plays an important role in the decline of hygrophilous Orchidaceae [55,56], especially for members of the genera *Dactylorhiza* and *Serapias*, which are threatened due to habitat loss and the drainage of wetlands. In the Talassemtane National Park, we observed during the 2023 survey the drying out of the local wetland (site 7) and a significant reduction in the abundance of *Serapias* spp. Moreover, previous reports highlight the existence of two species of the genus *Dactylorhiza* in the Talassemtane National Park [28], which we did not encounter during our extensive field studies; thus, they are not confirmed herein.

In general, the conservation of Orchidaceae in urban and/or peri-urban areas is only possible with the minimization of human pressures and/or under the accurate management of orchid habitats [26]. Concerning the present study, the growth sites of Orchidaceae in the Dardara (sites 33–37) and Jbel Lahbib (sites 38–42) areas constitute prime recreational areas; therefore, their wild-growing orchid populations are highly threatened, thus requiring urgent and rational interventions under prioritized schemes. The presence of numerous human activities in the study area could further facilitate the colonization success of some notoriously expansive species, such as *Rubus ulmifolius* and *Dittrichia viscosa*. Consequently, an increase in *R. ulmifolius* cover could lead to a reduction in the surface area occupied by herbaceous plants such as orchids [57], while the expansion of *D. viscosa* could cause economic and environmental damage, excluding other native plants [58].

Natural factors, such as substrate, exposure, and/or moisture, could influence Orchidaceae diversity and distribution in the wild. It has been reported that compared to limestone habitats, the serpentine bedrock habitat could also be suitable for orchid growth because of the open vegetation with generally lower levels of competition between plants [41]. However, only seven Orchidaceae taxa were found in the siliceous bedrock part of the Bouhachem Natural Park. This finding could be explained by the specific physico-chemical properties of the serpentine soils [41]. To overcome the problems of soil dryness and interspecific competition, the Orchidaceae of the study area probably tend to occupy habitats oriented towards less sunny exposures and spaces left open by the local vegetation and/or along the edges of roads and trails. Such orchid distribution patterns of light-demanding orchids associated with low competitive ability [41] could be explained by their preference for the relatively coldest slopes at the local scale, which are usually oriented toward the north and east [59].

Some orchids, such as *Ophrys scolopax,* are very popular in Spain and elsewhere due to their uses in folk traditions, in which they have special agro-alimentary value for children and convalescents [17]. In Morocco, however, no such knowledge appears among the local inhabitants (e.g., [60,61,62]). During this study, we encountered many people during fieldwork who did not recognize orchids or their uses. Such plants were referred to by some local people as ‘Bossila’, a local common name for bulbous plants in general. However, among experienced herbalists, local orchid bulbs are sold for medicinal purposes [18]. One way or another, the lack of knowledge and the possible uncontrolled trade could threaten wild orchids in the study area.

The protection of orchid populations through the protection of specific areas could contribute to their conservation [63,64]. However, this is not always the case. As reported by Khapugin [26], human activities may well persist, even in protected areas, especially when near urbanized zones. Therefore, for the conservation of orchids in the IBRM, the enforcement of existing laws, such as Law No. 29-05 of 2011 on the protection of species of wild fauna and flora and their trade, should be combined with and supplemented by other measures and initiatives, such as the creation of a network of legally protected areas and/or the expansion of the boundaries of existing ones, the development of profitable socioeconomic projects at the local scale, the adoption of appropriate tourism infrastructure, and raising awareness among the local population.

In parallel with the in situ conservation of Orchidaceae species, ex situ conservation represents another approach to preserving plant resources [52,65,66]. Such efforts should employ the conservation of viable plant material in botanical gardens and seed banks, as well as the promotion of scientific research related to species-specific germination trials of orchid seeds by testing various abiotic conditions likely to influence their germination, such as water stress, salinity, pH, etc. Such experimentation and applied research could develop expertise in raising ex situ new plant material that is less vulnerable to climatic hazards, with the aim of reinforcing declined or threatened wild-growing populations through their reintroduction into natural habitats [62,66,67,68].

## 5. Conclusions

The study of Orchidaceae diversity in the Moroccan part of the Intercontinental Biosphere Reserve of the Mediterranean identified a remarkable biodiversity of 21 taxa in seven Orchidaceae genera, thus increasing the knowledge about these plants in Morocco (new national records, new local records, and new occurrences). The presence of natural plant formations such as fir forests, cork oak forests, holm oak forests, and zeen oak forests, as well as semi-anthropized formations like matorrals dominated by mastic trees within this biodiversity reserve, provides a variety of favorable habitats for Orchidaceae members. However, these habitats are threatened by climate change and human activities such as water drainage, an increase in fire frequency/intensity, *Cannabis* farming, etc. Due to numerous natural threats and human pressures, several Orchidaceae species may disappear from the wild habitats of the IBRM soon. It is, therefore, recommended to consider the preservation of Orchidaceae habitats when planning future land use and socioeconomic development projects at local or regional scales. The information collected during this study could be useful as a baseline for raising awareness among the local population and decision-makers to take appropriate measures for the in situ conservation of Orchidaceae. These measures include the monitoring and constant surveillance of existing wild-growing populations based on the species-specific and site-specific counting herein reported for the first time, as well as the adoption of proper habitat management schemes (prohibition of wetland drainage, control of *Cannabis* farming, grazing control, and appropriate forest management). Our study could constitute an initiative to launch an integrated conservation process of Orchidaceae taxa in Morocco. This process requires more work at the national level to identify trends, threats, and conservation priorities on the local scale, which should be aligned with wider conservation strategies on a global scale.

## Figures and Tables

**Figure 1 plants-14-01254-f001:**
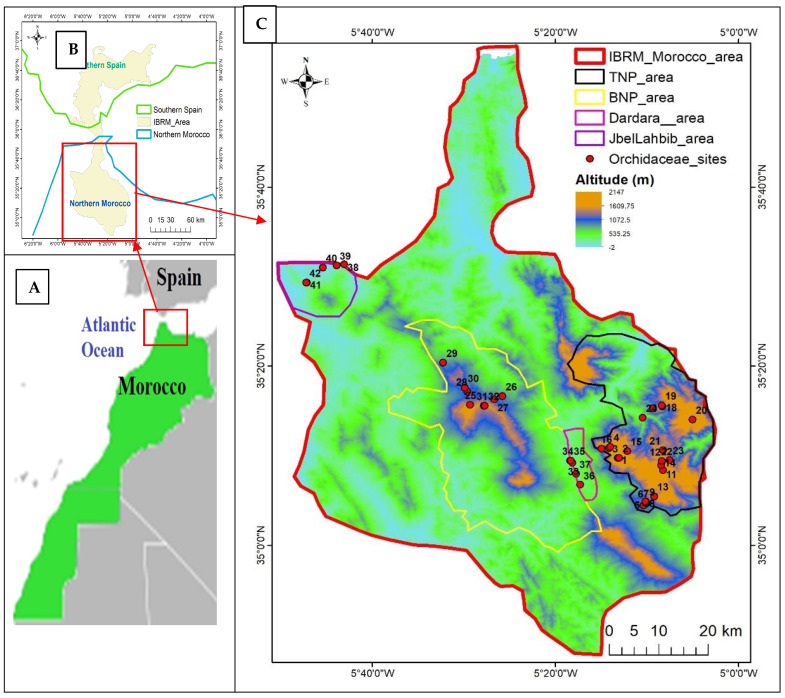
Map of the Intercontinental Biosphere Reserve of the Mediterranean (IBRM) shared between Spain and Morocco (**A**) showing the localization of the Moroccan study area (**B**) and the investigated sites in the Moroccan part of the IBRM (**C**), including four distinct protected areas, namely the Talassemtane National Park (TNP: sites 1–24), the Bouhachem Natural Park (BNP: sites 25–32), the Dardara area (sites 33–37), and the Jbel Lahbib Reserve (sites 38–42).

**Figure 2 plants-14-01254-f002:**
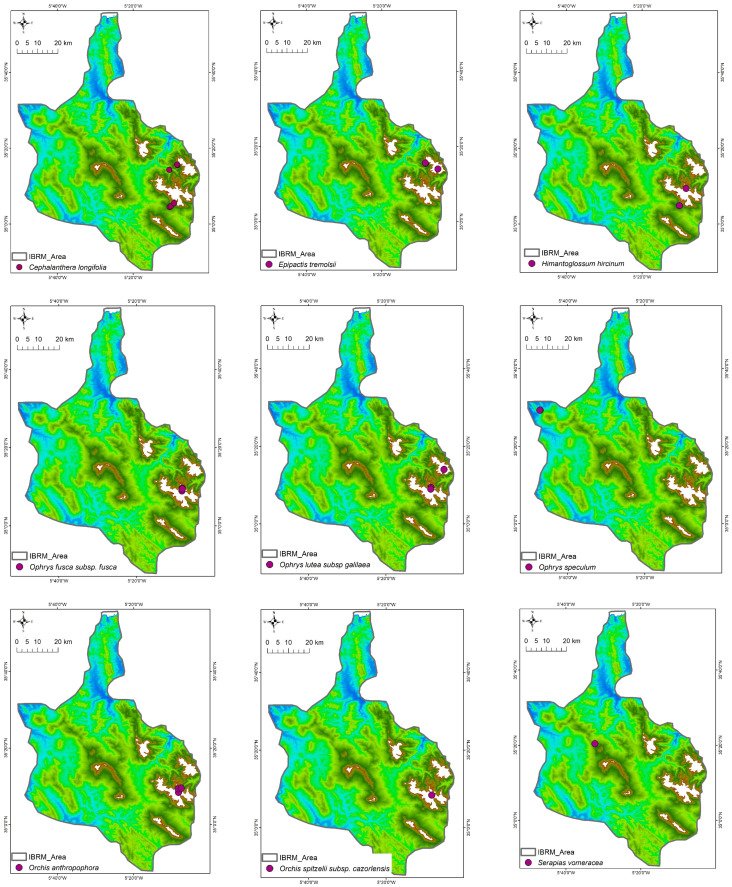
Distribution patterns of orchid taxa that were recorded in only one of the protected areas studied (*n =* 9).

**Figure 3 plants-14-01254-f003:**
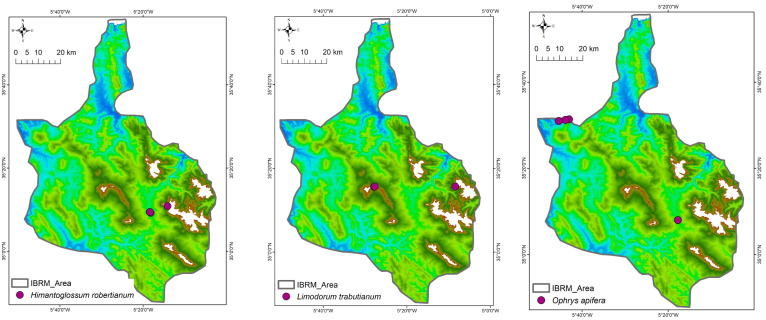
Distribution patterns of orchid taxa that were recorded in two of the protected areas studied (*n =* 10).

**Figure 4 plants-14-01254-f004:**
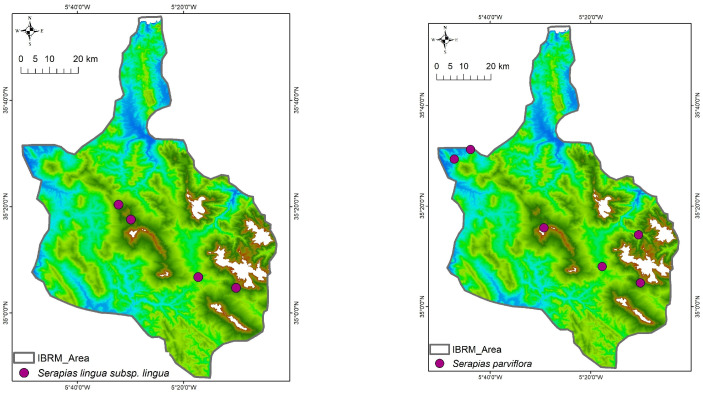
Distribution patterns of *Serapias* taxa that were recorded in three or all protected areas studied.

**Figure 5 plants-14-01254-f005:**
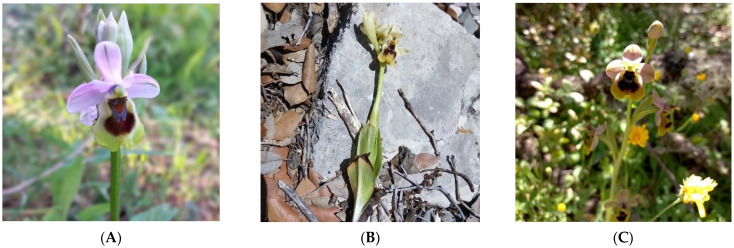
Variation of *Ophrys tenthredinifera* presumed subspecies according to the African Plant Database [46] in the studied Moroccan part of the Intercontinental Biosphere Reserve of the Mediterranean: (**A**) *Ophrys tenthredinifera* Willd. subsp. *tenthredinifera* (Talassemtane National Park, 24 April 2021); (**B**) *Ophrys tenthredinifera* subsp. *grandiflora* (Ten.) Kreutz (Talassemtane National Park, 1 May 2019); (**C**) *Ophrys tenthredinifera* subsp. *ficalhoana* (J.A.Guim.) M.R.Lowe & D.Tyteca (Talassemtane National Park, 19 May 2022).

**Figure 6 plants-14-01254-f006:**
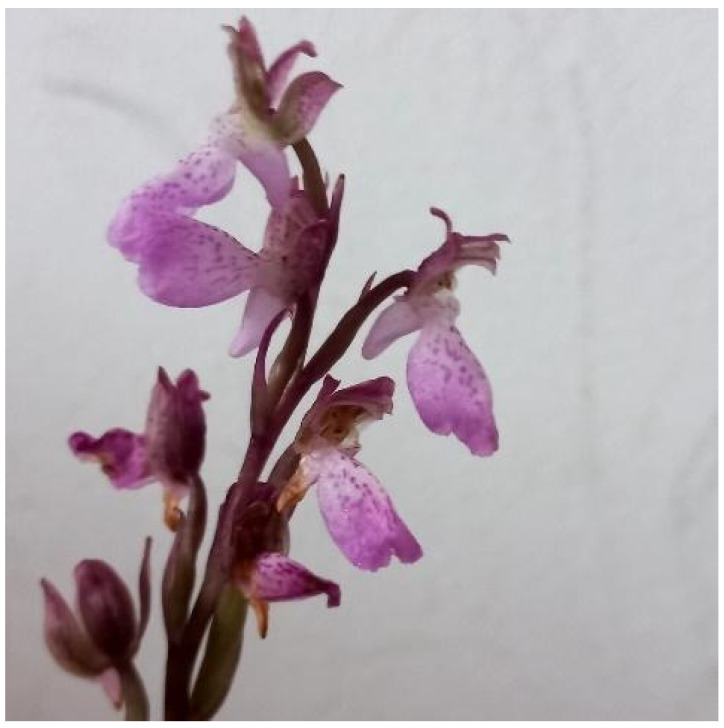
*Orchis spitzelii* subsp. *cazorlensis,* a newly recorded member of Orchidaceae in the studied Moroccan part of the Intercontinental Biosphere Reserve of the Mediterranean (Talassemtane National Park, 19/05/2022).

**Figure 7 plants-14-01254-f007:**
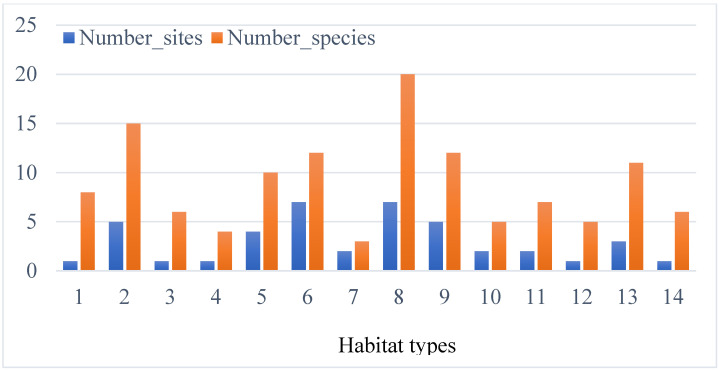
Richness of non-Orchidaceae vascular plants per habitat type in the Moroccan part of the International Biosphere Reserve of the Mediterranean. Habitat types: 1. Chamaephyte-rich with *Chamaerops humilis* and *Stachys fontqueri*; 2. Coniferous rainforest with forest *Abies pinsapo* subsp. *marocana*; 3. Matorral with *Pistacia atlantica*, *Olea europaea* subsp. *europaea* and *Chamaerops humilis*; 4. Matorral with *Pistacia lentiscus* and *Chamaerops humilis*; 5. Matorral with *Pistacia lentiscus*, *Myrtus communis,* and *Olea europaea* subsp. *europaea*; 6. Matorral with *Quercus ilex*; 7. Pelouses with Asteraceae, Boraginaceae, and Poaceae members; 8. Rainforest with *Quercus ilex* and *Cistus* spp.; 9. Rainforest with *Quercus suber, Pistasia lentiscus*, and *Myrtus communis*; 10. Rainforest degraded on siliceous substrate with *Pteridium aquilinum*; 11. Rainforest on siliceous substrate with *Quercus canariensis, Quercus lusitanica,* and *Quercus pyrenaica*; 12. Rainforest on siliceous substrate with *Quercus pyrenaica* and *Quercus suber*; 13. Rainforest on siliceous substrate with *Quercus suber* and *Quercus pyrenaica*; 14. Rainforest with *Quercus ilex* and *Quercus faginea*.

**Figure 8 plants-14-01254-f008:**
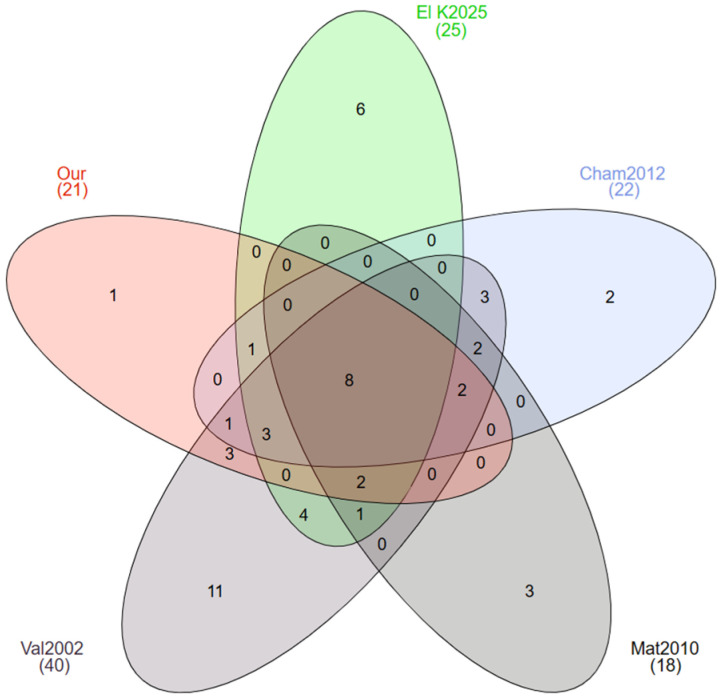
Venn diagram showing the number of common and non-common orchid taxa (species and subspecies) between the current (our) and previous studies in the Rif region, Morocco. Sources: Val2002-[30]; Mat2010-[28]; Cham2012-[27]; El K2025-[14].

**Table 1 plants-14-01254-t001:** Habitat types identified in the study area.

No	Habitat Types
1	Chamaephytic vegetation with *Chamaerops humilis* and *Stachys fontqueri*
2	Coniferous forest with *Abies pinsapo* subsp. *marocana*
3	Matorral with *Pistacia atlantica*, *Olea europaea* subsp. *europaea*, and *Chamaerops humilis*
4	Matorral with *Pistacia lentiscus*, and *Chamaerops humilis*
5	Matorral with *Pistacia lentiscus*, *Myrtus communis*, and *Olea europaea* subsp. *europaea*
6	Matorral with *Quercus ilex*
7	Pelouses with Asteraceae, Boraginaceae, and Poaceae members
8	Rainforest with *Quercus ilex*
9	Rainforest with *Quercus suber*, *Pistasia lentiscus*, and *Myrtus communis*
10	Rainforest degraded on siliceous substrate with *Pteridium aquilinum*
11	Rainforest on siliceous substrate with *Quercus canariensis*, *Quercus lusitanica*, and *Quercus pyrenaica*
12	Rainforest on siliceous substrate with *Quercus pyrenaica* and *Quercus suber*
13	Rainforest on siliceous substrate with *Quercus suber* and *Quercus pyrenaica*
14	Rainforest with *Quercus ilex* and *Quercus faginea*

**Table 2 plants-14-01254-t002:** Collection data for Orchidaceae taxa (species and subspecies) in the surveyed sites of the studied Moroccan part of the Intercontinental Biosphere Reserve of the Mediterranean. NT: number of taxa; NI: number of individuals; A: annual and perennial non-timber crops; AW: abstraction of surface water (agricultural use); F: increase in fire frequency/intensity; GP: gathering terrestrial plants; R: roads; G: small-holder grazing, ranching, or farming; T: tourism and recreation areas; W: solid waste.

Area	Site	Date	Latitude (N°)	Longitude (W°)	Elevation (m)	NT	NI	Vegetation Cover (%)	Threat Types
Talassemtane National Park	1	1 May 2019	35.1632	−5.2199	1876	1	3	75–100	G
	2	1 May 2019	35.1625	−5.2178	1969	1	20	75–100	G
	3	12 April 2021 24 April 2021	35.1808	−5.2355	1147	1	2	75–100	A, R
	4	12 April 2021 24 April 2021	35.1833	−5.2338	1219	1	12	50–75	G, R, T, W
	5	24 April 2021	35.0741	−5.1725	1134	1	1	25–50	-
	6	19 May 2022	35.0782	−5.1676	1156	3	25	75–100	R, W
	7	24 April 2021 19 May 2022	35.079	−5.1689	1149	3	80	75–100	AW, R, W
	8	24 April 2021	35.0802	−5.1688	1174	2	14	50–75	R
	9	24 April 2021	35.0819	−5.1694	1175	1	15	75–100	R
	10	24 April 2021 19 May 2022	35.0819	−5.1697	1216	1	42	75–100	GP
	11	19 May 2022	35.1397	−5.1375	1715	3	37	75–100	-
	12	24 April 2021 19 May 2022	35.1556	−5.1404	1608	6	73	50–75	G
	13	19 May 2022 26 April 2023	35.0911	−5.1536	1294	5	63	75–100	G
	14	19 May 2022	35.1492	−5.1414	1664	4	28	25–50	-
	15	19 May 2022	35.1752	−5.203	1347	1	2	75–100	-
	16	19 May 2022	35.1797	−5.2502	1026	1	1	75–100	-
	17	26 April 2023	35.2553	−5.1558	1112	1	2	75–100	R
	18	26 April 2023	35.2585	−5.1392	1366	2	41	50–75	-
	19	26 April 2023	35.2605	−5.1402	1318	4	61	75–100	-
	20	26 April 2023	35.234	−5.0841	1573	4	37	75–100	AF
	21	26 April 2023	35.1771	−5.1382	1364	1	8	75–100	-
	22	26 April 2023	35.1565	−5.1403	1600	2	14	50–75	-
	23	26 April 2023	35.1586	−5.1261	1352	2	34	50–75	-
	24	18 May 2024	35.2378	−5.1749	415	2	7	50%-75	T
Bouhachem Natural Park	25	11 March2023 11 May 2024	35.2614	−5.4889	1293	1	11	75–100	G, T
	26	11 May 2024	35.2773	−5.4301	849	1	12	75–100	G, T
	27	11 May 2024	35.2721	−5.4447	1010	1	6	75–100	G, R, T, W
	28	11 May 2024	35.2861	−5.494	1113	1	10	50–75	G
	29	11 May 2024	35.3398	−5.538	844	2	26	50–75	G
	30	11 May 2024	35.293	−5.4993	1071	2	24	75–100%	G
	31	11 March 2023 11 May 2024	35.2596	−5.4638	1471	2	18	50–75	G
	32	11 March 2023 11 May 2024	35.2603	−5.4623	1522	2	21	50–75	G
Dardara	33	5 March 2023	35.1537	−5.3033	306	1	14	25–50	R, W, T
	34	5 March 2023	35.1574	−5.3069	349	1	120	50–75	R, T
	35	5 March 2023	35.1575	−5.3065	330	1	30	25–50	R, A, T
	36	26 April 2023	35.1126	−5.2885	373	2	24	75–100	G, R, T, W
	37	26 April 2023	35.1329	−5.2957	330	3	57	50–75	G, R, T, W
Jbel Lahbib Reserve	38	9 March 2023 9 April 2023	35.5229	−5.7181	77	1	23	50–75	R, G
	39	9 March 2023 9 April 2023	35.5207	−5.7319	77	2	23	25–50	R, G
	40	9 March 2023 9 April 2023	35.5167	−5.757	88	1	21	50–75	R, G
	41	9 March 2023 9 April 2023	35.4887	−5.7862	107	5	47	75–100	R, G
	42	9 March 2023 9 April 2023	35.4886	−5.7872	104	1	29	25–50	R, G

**Table 3 plants-14-01254-t003:** Orchidaceae taxa (species and subspecies) and number of individuals recorded in different areas and sites in the studied Moroccan part of the Intercontinental Biosphere Reserve of the Mediterranean (nomenclature according to POWO [45]).

Taxa	Total Areas	Total Sites	Total Individuals
*Cephalanthera longifolia*	1	6	54
*Epipactis tremolsii*	1	3	36
*Himantoglossum hircinum*	1	2	5
*Himantoglossum robertianum*	2	4	166
*Limodorum trabutianum*	2	3	8
*Neotinea maculata*	2	6	63
*Ophrys apifera*	2	4	85
*Ophrys* x *battandieri*	2	6	69
*Ophrys fusca* subsp. *fusca*	1	3	10
*Ophrys lutea* subsp. *galilaea*	1	3	21
*Ophrys scolopax* subsp. *apiformis*	2	2	9
*Ophrys speculum*	1	2	32
*Ophrys tenthredinifera*	2	9	151
*Orchis anthropophora*	1	3	66
*Orchis mascula*	2	3	27
*Orchis mascula* subsp. *laxifloriformis*	2	9	156
*Orchis spitzelii* subsp. *cazorlensis*	1	1	13
*Serapias lingua* subsp. *lingua*	3	4	60
*Serapias parviflora*	4	5	76
*Serapias strictiflora*	2	2	19
*Serapias vomeracea*	1	1	12

## Data Availability

All floristic and ecological data obtained during the inventory are included in this study. The original dataset used in the analysis is available upon request.

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
