# Peer review of "In Situ Conservation of Orchidaceae Diversity in the Intercontinental Biosphere Reserve of the Mediterranean (Moroccan Part)"

_plants, 2025, doi:10.3390/plants14081254_

Round 1

Reviewer 1 Report

Comments and Suggestions for Authors

The manuscript contains valuable work on the orchids of northern Morocco and the factors that threaten them.

The taxonomy of orchids is highly divisive, so it is indeed important that nomenclature follows a single source and I recommend Plants of World Online, which is cited by the authors [37], although not consistently applied.

A presentation of the orchid flora of the region in the introduction, setting the context for the reader, would greatly enhance the value of the manuscript.

My detailed comments can be found in the attached file.

Author Response

REVIEWER 1 - Comments and Suggestions for Authors

The manuscript contains valuable work on the orchids of northern Morocco and the factors that threaten them.

Authors’ response: We would like to thank the reviewer for his/her kind comments.

The taxonomy of orchids is highly divisive, so it is indeed important that nomenclature follows a single source and I recommend Plants of World Online, which is cited by the authors [45], although not consistently applied.

Authors’ response: We have corrected the nomenclature of taxa appearing in the abstract as advised by the reviewer and we have only used the Plants of the World Online (POWO) approach. Within the manuscript’s texts, we have consistently applied the POWO’s nomenclature as a principal source; however, we have also indicated in brackets the nomenclature of the surveyed taxa as adopted by the African Plant Database which is the principal source for the African continent. The extant discrepancies are due to disagreement between these major nomenclatural sources; this represents an important issue that is discussed in our study similarly to other studies, see for example the following diagram as reported by Libiad et al. 2020:

Venn diagram showing the numbers (percentages in parentheses) of common taxa (species and subspecies) that are considered as endemic to the Mediterranean coast and Rif region of northern Morocco according to at least one of the four independent sources consulted in this study, i.e. Euro+Med Plantbase (E+M, http://www.emplantbase.org/home.html), The Plant List (PL, http://www.theplantlist.org/), African Plant Database (APD, http://www.ville-ge.ch/musinfo/bd/cjb/africa/ recherche.php) and Plants of the World Online (PWO, http://www.plantsoftheworldonline.org/), all accessed 13 Oct. 2019.

A presentation of the orchid flora of the region in the introduction, setting the context for the reader, would greatly enhance the value of the manuscript.

Authors’ response: We would like to thank you for this comment. To address the reviewer’s comment, we have further elaborated the introduction part in the revised version of the manuscript providing more data regarding current research on the orchid flora of the study region (see track changes).

My detailed comments can be found in the attached file.

Authors’ response: Each comment made by the reviewer 1 in the pdf file attached is addressed separately.

Comments appearing in the attached pdf file

ABSTRACT: Synonym of Orchis spitzelii subsp. cazorlensis according to PoWO [45] and Synonym of Ophrys fusca subsp. fusca according to PoWO [45]

Authors’ response, line 26:  We have corrected these names in the revised version of the manuscript following the POWO’s nomenclature in the abstract (see track changes).

Page 2:

Comment 3: This information is not relevant to the topic. Orchids are so specific and diverse that the authors should select from the most relevant characteristics

Authors’ response, line 43: We have deleted the redundant information in the revised version of the manuscript (see track changes).

Comment 4: It would be nicer if the authors focused on their area of research in a broader sense: the Mediterranean area, especially North Africa.

Authors’ response, line 49: Following the reviewer’s suggestion, we have elaborated this part in the introduction of the revised manuscript, so one may read it as follows: “Previous studies from North Africa have reported on the richness and diversity of Orchidaceae members, their distribution and their in-situ conservation [11-13]. These works have highlighted the large diversity of Orchidaceae in North Africa and emphasized the need for additional prospection and protection regarding certain taxa and areas [11-13]. Despite their ecological importance, Orchidaceae members of the Mediterranean basin, and their habitats are threatened by several human activities, namely annual crops, fire, grazing, over-collection, recreational activities and urbanization [6, 14-17]. Moreover, previous studies ring the bell that the presence of more than 135 Orchidaceae taxa (species and subspecies) is threatened by legal and/or illegal trade [17-22].”

Comment 5: For non-Moroccan readers, a more in-depth explanation of why these areas were selected is needed. These names, unfamiliar to us, have been picked out of thin air.

Authors’ response, line 72: These areas are nationally protected by Moroccan law despite severe knowledge gaps regarding their flora, and especially orchids. The initial scope therefore was to determine whether orchid members are conserved in these protected areas or not. Having reformulated the objectives of the study in the revised version of the manuscript as questions (see track changes at the end of the introduction part), this point has been made clear.

Comment 6: It would also be useful to outline the main characteristics of the orchid flora in the surrounding areas and their main threats. This would provide the reader with a basis for the results.

Authors’ response, line 79:  To address the reviewer’s comment in the suggested direction, we have elaborated this part in the introduction of the revised manuscript, so one may read the following new texts (see track changes):

- “Regionally, 20 orchids of Spain are included in the IUCN Red List including one threatened (EN) and three near threatened (NT) taxa; in contrast, however, only 13 taxa are included in the IUCN Red List and seven taxa are assessed as threatened in North Africa. In Morocco, only three taxa, namely Dactylorhiza maurusia (Emb. & Maire) Holub (EN), Dactylorhiza elata (Poir.) Soó (NT) and Platanthera algeriensis Batt. & Trab. (NT) are assessed as threatened [22, 23]. The latter two orchid species share the same threat categories in Morocco and Spain.”

- “Nonetheless, severe knowledge gaps hinder the protection of Orchidaceae members in Morocco as compared to neighboring countries that have conducted large inventories of Orchidaceae members, e.g., in the Iberian Peninsula and the Balearic Islands with more than 130 taxa [32, 33], in Spain with 96 taxa [15], in North-East Algeria with 64 taxa [12], or in Tunisia with 50 orchid taxa [11]. Moreover, Spanish managers have gone further by including 76% of the domestic native orchids in some catalogue of legal protection, such as IUCN Red List or the catalogue of regional protection or the regional Spanish red list [15]. In Andalusia (Southern Spain) including the northern part of the IBRM, four Orchidaceae taxa are included in the regional catalogue (RC) of Andalusia (DEC 23/2012) [15]. These conservation efforts have led to a reduction in threats to Spain's orchids. This situation has facilitated the use of orchids as bioindicators in habitat management and species conservation [15, 34]. The situation in Algeria is no better than in Morocco, with only seven taxa protected under the Algerian law for the conservation of wild plant species of 2012 [35]. Due to the gaps of knowledge regarding the diversity and threats of Orchidaceae compared to neighboring countries, Morocco and other North African countries must accelerate the inventory speed regarding the conservation of its Orchidaceae members especially in the IBRM region to prevent their possible disappearance in the face of the ongoing biodiversity loss crisis.”

Page 3:

Comment 7: Figures should be numbered in the order in which they are referred to in the text. This will be Figure 1.

Authors’ response, line 92: We have corrected the order of figures; all figures were re-enumerated as they appear in the revised version of the manuscript. All amendments are evident with track changes.

Comment 8: not species?

Authors’ response, line 99: This number refers to the total of 1138 orchid individuals of all species and subspecies that were recorded in the study area.

Page 4

Comment 9 : It would be nice to mention for each taxon listed whether it was known from that locality or whether it is a new record.

Authors’ response, line 119:  In the revised version of the manuscript, the floristic catalogue has been transferred to the Supplementary materials as Appendix S1. To accommodate the reviewer’s suggestion, in the Appendix S1 we have added for each taxon and studied area whether it was recorded for the first time (see new record in the area).

Page 12:

Comment 10 : It would also be necessary to present the listed species according to which taxa have a narrow distribution. The conservation of the latter in northern Morocco is much more important than that of the widespread species.

Authors’ response, line 220: Following the reviewer’s suggestion, we have added in the revised version of the manuscript the following text: “In the study area, seven taxa could be considered as prioritized in conservation terms due to narrow distribution range combined with small number of individuals (Epipactis tremolsii, Himantoglossum hircinum, Limodorum trabutianum, Ophrys fusca subsp. fusca, Orchis mascula, Orchis spitzelii subsp. cazorlensis, and Serapias vomeracea), thus requiring in situ conservation monitoring (Table 3). Novelty-wise, this inventory identified one new Orchidaceae taxon for the first time in Morocco, namely Orchis spitzelii subsp. cazorlensis (Figure 6).”

Page 15 :

Comment 11 : I consider the identification key in the manuscript unfortunate, as the taxonomy of Ophrys is indeed very complicated, almost all authors/databases have different opinions. This key should be supported by measurements and data, which has not been done.

Authors’ response, line 391: As suggested by the reviewer, the identification key has been deleted from the revised version of the manuscript.

Page 16:

Comment 12: According to the methodology, no taxonomic studies have been undertaken, so I do not consider the details of species descriptions to be relevant.

Authors’ response, line 432: As suggested by the reviewer, the species descriptions have been deleted from the revised version of the manuscript.

Page 19:

Comment 13: It means a dry and warm summer? Why do the authors keep repeating 'summer'? Is there a special reason?

Authors’ response, line 568: The repetition was deleted in the revised version of the manuscript.

Comment 14: see the previous comment

Authors’ response, line 569: The repetition was deleted in the revised version of the manuscript.

Comment 15: Was there any prior documentation or literature that you could draw on? Was there previous orchid data from these areas?

Authors’ response, line 586: Following the reviewer’s suggestion, previous knowledge and orchid data from these areas (Matéos et Valdès, 2010, Chambolyron 2012; Vazquez et al. 2012) have been added to the revised manuscript and discussed (see track changes).

Comment 16: How was the coverage and size of each site determined?

Authors’ response, line 599: The site coverage rate was determined by estimating the proportion of the area covered by vascular vegetation.
The site size was defined by the presence of a continuous population of a recorded orchid species. If the continuity of the orchid population was disrupted by a change in exposure, habitat, or altitude, then it was considered a different site.

Page 20 :

Comment 17 : I don't see this in the manuscript. The 2 taxon names cited in the abstract are also synonymous according to [45]. And so are many other cases. I recommend the authors really use the names under [45}.

Authors’ response, line 634: Following the reviewer’s suggestion, all scientific names of plants used in the manuscript follow primarily POWO [45].

Page 24:

Comment 18 : also previously mentioned

Authors’ response: Following the reviewer’s suggestion, all scientific names of plants used in the manuscript follow primarily POWO [45].

Reviewer 2 Report

Comments and Suggestions for Authors

Orchids are usually endangered and highly species differentiated groups, and there have important ecological protection and utilization value in regional biodiversity. The Dr. Karmoudi et al surveyed the Orchidaceae in an IBMR in Northern Morocco. The MS afford abundant field data of orchid, but the writing is terrible. Some suggest for you revise you MS. This is not a journal article for communicate with international readers, but a report for local government official or administrator.

  1. You should identify your research objectives and scientific questions in Introdution.
  2. You MS lacked the Materials and Methods, you should afford how did you get and analysis the data for you article.
  3. I suggest you to analysis the species diversity, distribution pattern, habitats characteristics, especially the endangered orchids.

Author Response

REVIEWER 2 - Comments and Suggestions for Authors

Orchids are usually endangered and highly species differentiated groups, and there have important ecological protection and utilization value in regional biodiversity. The Dr. Karmoudi et al surveyed the Orchidaceae in an IBMR in Northern Morocco. The MS afford abundant field data of orchid, but the writing is terrible. Some suggest for you revise you MS. This is not a journal article for communicate with international readers, but a report for local government official or administrator.

Authors’ response: Compared to the initial submission, the revised version of the manuscript: (i) has been restructured following editorial advice and comments; (ii) the annotated floristic catalogue has been transferred to a new Supplementary material (Appendix S1); (iii) the species distribution maps have been presented as combined plates (Figures 2, 3 and 4), (iv) new texts have been added, and (v) all texts have been extensively revised to correct grammar and linguistic imperfections (see track changes). We remain at the disposal of the reviewer for further specific improvements.

You should identify your research objectives and scientific questions in Introdution.

Authors’ response: Following the reviewer’s advice, we have elaborated the end of the introduction part and we have presented clear questions that are answered by the presented results.

You MS lacked the Materials and Methods, you should afford how did you get and analysis the data for you article.

Authors’ response: We have strictly followed the template of MDPI PLANTS; therefore, the Material and Methods section initially appeared after the Results and Discussion section and prior to the Conclusions section in the submitted manuscripts (the Material and Methods section was originally submitted). However, due to editorial advice we have transferred the Material and Methods section right after the introduction part.

I suggest you to analysis the species diversity, distribution pattern, habitats characteristics, especially the endangered orchids.

Authors’ response: Following the reviewer’s suggestion, we have added in the revised version of the manuscript the following text: “In the study area, seven taxa could be considered as prioritized in conservation terms due to narrow distribution range combined with small number of individuals (Epipactis tremolsii, Himantoglossum hircinum, Limodorum trabutianum, Ophrys fusca subsp. fusca, Orchis mascula, Orchis spitzelii subsp. cazorlensis, and Serapias vomeracea), thus requiring in situ conservation monitoring (Table 3). Novelty-wise, this inventory identified one new Orchidaceae taxon for the first time in Morocco, namely Orchis spitzelii subsp. cazorlensis (Figure 6).”

Reviewer 3 Report

Comments and Suggestions for Authors

All in the MS

Author Response

REVIEWER 3 - Comments and Suggestions for Authors

- In situ in italic:

Authors’ response: We have corrected this shortcoming each time it appears in the revised manuscript (see track changes). However, based on our experience with previous articles in MDPI journals, this change will most probably be restored to the initial one (in-situ not italicized).

- Keywords in alphabetic way

Authors’ response: We have arranged the keywords in alphabetical order as requested by the reviewer.

- Intercontinental Biosphere Reserve of the Mediterranean in abbreviation:

Authors’ response: We have corrected this in the revised manuscript (see track changes).

- Ethic aspects: The authors of the article ensured that field surveys do not affect the survival of the studied Orchidaceae populations.

Authors’ response: To address the reviewer’s concern, we have added in the revised version of the manuscript (see track changes, last paragraph of the transferred Materials and Methods): “In each site studied, the living (fresh) specimens of Orchidaceae species were in-situ identified and photographed without harming the wild-growing individuals and populations.”

Round 2

Reviewer 2 Report

Comments and Suggestions for Authors

The revised version has been greatly improved for writing, data analysis, tables and figures. But some minor problems remain in the MS. So, I suggest a few things below.

Title

  1. Revised the title to “Diversity and in-situ conservation of Orchidaceae in the the Intercontinental Biosphere Reserve of the Mediterranean of Moroccan part.

Abstract

Line24 delete this study reported the identification of.

Line25 after seven genera insert was identified.

Line 26 after in Morocco insert comma, replace different areas surveyed to sites records.

Line 28 delete members of IBRM

Materials and Methods

Figure 1, I suggest you to afford entire boundary of IBRM and identify IBRM of Moroccan part, which the Moroccan part is you study area.

Otherwise, I cannot understand the whiter color mean in the central of your study area, but I can read from blue to brown.

Line139 What is the spatial relationships among the IBRM Moroccan part, Talassemtane National Park and Bouhachem Natural Park? I suggest you to mark all key place on you map, which appeared in the Study Area, such as Dardara, Jbel Lahbib Reserve.

Table1 What were the differents among row 3-6, and row 11-12?

Results

Line229 What is APD mean in your MS?

Figure2, Figure 3 and Figure 4, Why didnot you plot all those species in one map with different identifier?

Comments on the Quality of English Language

I had revised some mistake in the PDF version as attachment as above.

Author Response

REVIEWER 2

The revised version has been greatly improved for writing, data analysis, tables and figures. But some minor problems remain in the MS. So, I suggest a few things below.

Title

  1. Revised the title to “Diversity and in-situ conservation of Orchidaceae in the the Intercontinental Biosphere Reserve of the Mediterranean of Moroccan part.

Authors’ response: We have rephrased the title of the revised manuscript to accommodate the suggestion made by the reviewer (see track changes). The new title is: Orchidaceae Diversity and In-Situ Conservation of the Inter-continental Biosphere Reserve of the Mediterranean (Moroccan part)

Abstract

Line24 delete this study reported the identification of.

Line25 after seven genera insert was identified.

Line 26 after in Morocco insert comma, replace different areas surveyed to sites records.

Line 28 delete members of IBRM.

Authors’ response: All suggested changes by the reviewer in the abstract have been incorporated in the revised version of the manuscript (see track changes).

Materials and Methods

Figure 1, I suggest you to afford entire boundary of IBRM and identify IBRM of Moroccan part, which the Moroccan part is you study area.

Otherwise, I cannot understand the whiter color mean in the central of your study area, but I can read from blue to brown.

Line139 What is the spatial relationships among the IBRM Moroccan part, Talassemtane National Park and Bouhachem Natural Park? I suggest you to mark all key place on you map, which appeared in the Study Area, such as Dardara, Jbel Lahbib Reserve. Talassemtane National Park, Bouhachem Natural Park, Dardara and Jbel Lahbib Reserve are parts of the IBRM (see new map, figure1)

Authors’ response: Following the reviewer’s advice, the white color was changed in Figure 1, and the surveyed study areas with enumerated sites have been outlined with different colors which are explained in the legend (see new Figure 1).

Table1 What were the differents among row 3-6, and row 11-12?

Authors’ response: Table 1 presents the different habitat types identified in the surveyed areas based on the dominant species prevailing in the local plant communities (see amendments with track changes).

Results

Line229 What is APD mean in your MS?

Authors’ response: APD means the African Plant Database specialist website (https://africanplantdatabase.ch). This has been clearly indicated in the revised version of the manuscript (see line 230).

Figure2, Figure 3 and Figure 4, Why didnot you plot all those species in one map with different identifier?

Authors’ response: The mapped Orchidaceae members were clustered in groups based on their distribution patterns. Furthermore, we preferred to map individually each Orchidaceae taxon for future monitoring purposes, showing at the same time true absences of their occurrence in the surveyed sites.
